# Survey of Scientific Rigor Studied in Machine Learning

## Abstract

The concern that Artificial Intelligence (AI) and Machine Learning (ML) are entering a "reproducibility crisis" has spurred significant research in the past few years. Yet with each paper, it is often unclear what someone means by "reproducibility" and where it fits in the larger scope of what we will call the "scientific rigor" literature. Ultimately, the lack of clear rigor standards can affect the manner in which businesses seeking to adopt AI/ML implement such capabilities. In this survey, we will use 66 papers published since 2017 to construct a proposed set of 8 high-level categories of scientific rigor, what they are, and the history of work conducted in each. Our proposal is that these eight rigor types are not mutually exclusive and present a model for how they influence each other. To encourage more to study these questions, we map these rigors to the adoption process in real-world business use cases. In doing so, we can quantify gaps in the literature that suggest an under focus on the issues necessary for scientific rigor research to transition to practice.

## 1 Introduction

The Artificial Intelligence (AI) and Machine Learning (ML) communities are becoming increasingly concerned with the "reproducibility" of their fields. This has come on the heels of a reproducibility crisis noted in many others. We will refer to this overarching concern, that the science of research is being done with some error rate, as a generic "scientific rigor" concern. This concern is justified, and increasingly it is challenging to evaluate the state of research around scientific rigor due to confused and incompatible usage of the same few terms like "reproducibility" (Plesser, 2018).

Due to confusing and often inconsistently used terminology in the literature, it is challenging to understand precisely what issues of scientific rigor the community is tackling, and how these issues may affect businesses seeking to integrate AI/ML into their operational capabilities. In light of these issues, we propose a new formulation of current scientific rigor research by surveying the current papers by the topics they cover. Further, we recognize AI and ML are increasingly used by other fields and becoming critical to many business operations. For this reason, we will further ground the discussion of our results in the context of adoption by a business process to solve some real-world problem. We find that this conjoint analysis allows us to better describe the scope of scientific rigor research and identify critically under-served areas that will have practical and real-world benefits in easier and faster adoption into practice. We hope this provides a tangible incentive for others to justify research in scientific rigor as having a real-world downstream purpose and impact.

In this survey, we will expand the ACM's proposed terminology of Repeatability, Reproducibility, and Replicability which we find useful, although still insufficient to capture the breadth of works done to date. Our contribution categorized current AI/ML research on scientific rigor into eight aspects we label as *repeatability*, *reproducibility*, *replicability*, *adaptability*, *model selection*, *label/data quality*, *meta & incentive*, and *maintainability*. These eight aspects are defined in Table 1. We propose these aspects based on our review of 66 papers published since 2017 and reflect the focus of the community at large. Table 1 also shows for each aspect the proportion of papers focused primarily on that aspect (though many papers touch on multiple aspects).

The rest of this article is organized as follows. First, we will detail the procedure we used to generate a set of 66 articles for analysis in section 2. The eight topics are derived from the exhaustive reading of the

Table 1: Eight primary topics that have been collectively described as "reproducibility" in the literature, determined by our manual review. The first three are based on the ACM's guidelines, the rest are informed by surveying and categorizing the themes of existing literature.

| Topics | Main Concern | % of papers |
|---|---|---|
| Repeatability | Can the results be obtained by the original authors using their original code and data. | 16.9 |
| Reproducibility | Can a different team obtain the results using the code and data provided by the original authors | 15.4 |
| Replicability | Can a different team, using different code and/or data, obtain the same results or results congruent with the original publication. | 6.2 |
| Adaptability | Can the original authors using the original code obtain qualitatively similar results on new/different data. | 4.6 |
| Model Selection | Given a set of two or more models, what process can be used to meaningfully and reliably determine which model to select for use. | 27.7 |
| Label/Data Quality | Given a process for labeling data, how can we ensure that the process results in meaningfully same labels over time and that the process of labeling has minimal errors. | 6.2 |
| Meta & Incentive | What are the motivators, or lack thereof, for scientific rigor. | 9.2 |
| Maintainability | What are the issues and remediations in running the same AI/ML solution as the people, code, and data are all altered in their nature over time. | 13.8 |

articles selected and inferred from themes and consistencies observed over the selected papers. Next, we will summarize the eight major topic areas of scientific rigor in section 3, with sub-areas included based on our literature review. Based on this survey of the literature we propose relationships for how these rigors interact in section 4, that we find informative as a macro-level picture of the scope of scientific rigor. In performing this survey, we find that it is often repeated that work of this type is not sufficiently rewarded or incentivized. For this reason we propose a stylized analysis of how new AI/ML methods come to be adopted in industry in section 5. This analysis suggests both motivation and justification for each type of rigor, as well as gaps between the amount of research done today and the importance with respect to the adoption process. Finally, we conclude in section 6.

## 2 Developing the Scope

Scientific rigor as a target of study is new within the machine learning field, and as of today, there are no dedicated venues to the topic. This makes it challenging to perform a collection of the literature to survey and summarize. For this reason, we will detail how we came to a list of works to survey as the core papers to summarize into distinct topics.

First, we started with a list of seminal papers known to us within the AI and ML fields broadly. Other notable works on reproducibility that are outside the field of machine learning are, such as (Ioannidis, 2005a), were excluded from this list because we are focused on machine learning specifically. Our starting list of papers was thus:

1. Odd Erik Gundersen and Sigbjørn Kjensmo. State of the Art: Reproducibility in Artificial Intelligence. *Proceedings of the 32nd AAAI Conference on Artificial Intelligence (AAAI-18)*, pp. 1644–1651, 2018

2. Edward Raff. A Step Toward Quantifying Independently Reproducible Machine Learning Research. In *NeurIPS*, 2019. URL `http://arxiv.org/abs/1909.06674`. arXiv: 1909.06674

3. Xavier Bouthillier, César Laurent, and Pascal Vincent. Unreproducible Research is Reproducible. In *Proceedings of the 36th International Conference on Machine Learning*, volume 97, pp. 725–734.

PMLR, 2019. URL `http://proceedings.mlr.press/v97/bouthillier19a.html`. Series Title: Proceedings of Machine Learning Research

4. D Sculley, Gary Holt, Daniel Golovin, Eugene Davydov, Todd Phillips, Dietmar Ebner, Vinay Chaudhary, Michael Young, Jean-Francois Crespo, and Dan Dennison. Hidden Technical Debt in Machine Learning Systems. In *Proceedings of the 28th International Conference on Neural Information Processing Systems - Volume 2*, pp. 2503–2511, Cambridge, MA, USA, 2015. MIT Press. Series Title: NIPS'15

5. Joelle Pineau, Philippe Vincent-Lamarre, Koustuv Sinha, Vincent Larivière, Alina Beygelzimer, Florence D'Alché-Buc, Emily Fox, and Hugo Larochelle. Improving Reproducibility in Machine Learning Research (A Report from the NeurIPS 2019 Reproducibility Program). *arXiv*, 2020. URL `http://arxiv.org/abs/2003.12206`. arXiv: 2003.12206

6. Chris Drummond. Replicability is not reproducibility: nor is it good science. In *Proceedings of the Evaluation Methods for Machine Learning Workshop at the 26th ICML, Montreal, Canada,2009*, 2009. Series Title: Evaluation Methods for Machine Learning Workshop, the 26th ICML, June 14-18, 2009, Montreal, Canada

From this initial seed set of papers, we reviewed all citations to each paper, as noted by Google Scholar, to identify any paper that seemed to be relevant to machine-learning and scientific rigor by subjective exhaustive review. Each paper that was identified as being relevant to scientific rigor was used to repeat the search for citations to/from the paper, and the Google Scholar profile of the lead author was reviewed for additional papers of relevance. Any venue that was discovered in the survey process that was focused on scientific rigor of some form (i.e., workshops) was also reviewed.

The cumulative citations of this article represent all of the found works we deemed relevant to the question of scientific rigor. However, the final list we used to read in exhaustive detail to synthesize core topics required further refinement to keep the scope reasonable. Thus the following filters were applied:

1. The paper had to be put on arXiv or published between 2017 and 2022, so that the survey would reflect what current researchers are focusing on.

2. The paper had to present itself as being about "reproducibility" or scientific rigor as a core theme/concept of the paper. Papers that tackled the same issue, but from other motivations, were excluded. This choice was made to try and have the survey reflect what those focusing on scientific rigor are considering.

The final set of 66 papers can be found in Appendix A.

## 3 The Current Scope of Work

Our literature survey identifies at least eight primary aspects of scientific rigor studied in the AI/ML literature. Each major sub-section will repeat one of the eight rigors defined in Table 1, and include further delineation for nuanced sub-categories that are present or noteworthy in the literature.

Before we detail these aspects, it is worth noting that many extant articles are best summarized as opinion pieces with varying degrees of formalization of their arguments. Most of these articles propose strategies or arguments on how to obtain "reproducibility", without evidence of effect (Gundersen & Kjensmo, 2018; Matsui & Goya, 2022; Publio et al., 2018; Tatman et al., 2018). Others have provided opinion pieces on the merits of focusing on Reproducibility (Lin, 2022) vs Replicability (Drummond, 2009; 2018; Raff & Farris, 2022) as the target goal of the AI/ML communities from a scientific perspective. The contents of this survey are focused on works that study issues, incentives, or interventions to rigor issues in AI/ML In surveying the current literature, and so does not include these opinion articles. However, we find this and the preceding section 4 informative to a broader scope understanding that it is not a choice between Reproducibility or Replicability, but a larger collective that must support all aspects of the study.

### 3.1 Repeatability

Repeatability is concerned with the authors obtaining the same results using the original source code and data. Interesting questions in repeatability include how to develop code and systems that make it easy for the developer to keep track of how they came to their experimental results from an experimental design perspective (Gardner et al., 2018; Paganini & Forde, 2020). In Human-Computer Interaction (HCI) research, there has been significant study on the iterative development nature of computational notebooks (e.g., Jupyter) that are widely used in AI/ML development processes. These notebooks can be prone to many subtle code errors/issues due to their fluidity and out-of-order execution. Enhanced tools can ensure the exact execution sequence to generate a result (Head et al., 2019; Kery & Myers, 2018). Many simple factors like using a random-number seed (i.e., for a Pseudo-Random-Number-Generator (PRNG)) are important for obtaining instantaneous repeatability.

Other factors like software version conflicts are often thought to be factors of repeatability but often lead to conflation. For example, does capture software versioning via a container system lead to repeatability, or reproducibility? We argue it would be reproducibility as a higher-level concept in our categorization, which we will detail further in section 4. A second distinction we make is that of instantaneous repeatability, vs. repeatability over time. In this immediate section we consider instantaneous repeatability, where the question is how to ensure repeatable results as the software/algorithm is being developed, and we find that there is surprisingly little beyond the work noted in the prior paragraph. When time is added as a factor, we consider this to be distinguishable as the maintainability rigor that we will detail in subsection 3.9.

### 3.2 Reproducibility

Reproducibility alters repeatability by imposing that a different individual/team be able to produce the same results using the original source code and data. This is a high focus of the AI/ML community and incentivization of Open Source Software (OSS) by major conferences and paper submission questionnaires/guidelines. Current work can be divided into those that explore surface-level issues such as unquantified proposals or exact procedure reproductions, vs those that attempt to quantify or better understand why a reproduction does(not) work.

#### 3.2.1 Surface Reproducibility

Surface-level studies of reproducibility report on the scale of the reproducibility challenge without examining whether their attempts at improving Reproducibility work. The only study we are aware of found that 74% of code released by the broader scientific community (beyond AI/ML) ran without issue (Trisovic et al., 2022). Toward remediating this in machine learning, many have proposed techniques like Docker to try and capture the exact conditions to re-run the experiments (Forde et al., 2018a;b).

#### 3.2.2 Reproducibility In Depth

A major factor in Reproducibility, and the discovery of non-reproducible work, is errors in the original comparisons being made. A seminal example from Metric Learning it was found that papers had multiple changes occurring simultaneously in comparison to prior baselines (new layers like Batch-Norm, optimizers, etc.) beyond just the proposed metric learning changes, which produced misleadingly large effect sizes (Musgrave et al., 2020). Broadly, many other works have identified similar issues nuanced to the sub-domain being studied (Lu et al., 2023; Liu et al., 2020).

The lessons from Musgrave et al. (2020) have re-occurred in many works since, especially in the Information Retrieval (IR) community. Such work includes studies using similar baseline errors/lack of tuning (Rao et al., 2022), studies expanding the set of baselines against an overly broad prior conclusion (Huang et al., 2022; Wang et al., 2022), and studies demonstrating that decades-old methods are still competitive when given the chance to run on larger modern datasets (Liu et al., 2022).

### 3.3    Replicability

Replicability is concerned with a different person/team being able to produce qualitatively similar results from the original article by writing their own code, and potentially different data. The aspect of replicability is highly under-studied, likely due to the challenges this aspect presents. Replicability requires re-implementing a target method's code from scratch, which is a labor-intensive process. Notable work in this direction was done by (Raff, 2019), who attempted to re-implement 255 papers, and computed features to quantify what properties correlated with a replicable paper. A volunteer effort by ReproducedPapers.org is collecting some Replicability attempts (Yildiz et al., 2021) based on which a thorough study in IR has been performed (Wang et al., 2022). Lastly, a unique approach to the Replicability question was studied by (Ahn et al., 2022), which focuses on the difference between computational floating point precision and the underlying symbolic math. From this perspective, they are able to suggest conditions about what statements can be rigorously tested and concluded about the math based on floating-point errors that would accumulate and cause issues otherwise.

We are not aware of other work within AI/ML on replicability. This state of affairs is common to other (relatively) code-free disciplines such as medicine (Ioannidis, 2005b) economics (Camerer et al., 2016) social sciences (Camerer et al., 2018). In these disciplines, replication studies are necessary and representative because it is the least costly way to evaluate a result. Other aspects of scientific rigor have a significantly lower barrier to entry, largely because AI/ML has a large open-source culture.

### 3.4    Model Selection

Model selection deals with the common task of AI/ML papers: given two competing methods (one of which may be the paper's own proposal), how do we conclude which method is better? As the AI/ML literature has advanced significantly through the presentation of empirically "better" algorithms, it is not surprising that the majority of historical and current work has focused on the model selection question. This includes how to pick and evaluate criteria for deciding "better", how to build benchmarks for a problem, and the process for determining "better" given criteria in a statistically sound way.

### 3.5    Evaluation Criteria & Methodology

Before one can select the "better" of one or more methods, it is first necessary to determine how the quality of a method is determined. The scope of evaluation metrics and scores is larger than that of scientific rigor, and this survey is concerned with cases where an invalid or errant procedure was identified and remediated. The literature in this direction is old, starting in the late 90s on the various pros-and-cons of metrics like Area Under the Curve (AUC) for evaluation (Bradley, 1997; Hand, 2009; Lobo et al., 2008). Likewise, work has addressed issues in scoring from leaderboards (Blum & Hardt, 2015), and subtle issues in using cross-validation to produce test-scores (Varma & Simon, 2006; Bergmeir et al., 2018; Varoquaux, 2018; Bates et al., 2021). Niche examples of the evaluation concern also exist. For example, three decades of malware detection performed subtle train/test leakage by adjusting for a target false-positive rate incorrectly (Nguyen et al., 2021) and time series anomaly detection scores being overly generous to "near hits" (Kim et al., 2022).

#### 3.5.1    Building Problem-Specific Benchmark Suites

It is becoming increasingly popular to build Benchmarks of multiple datasets, pre-prepared evaluation code, and methodology for specific problem domains (Blalock et al., 2020; Eggensperger et al., 2021; Sun et al., 2020). Such benchmark construction is popular, though yet has little agreed-upon practice with limited study at a macro level (Koch et al., 2021).

#### 3.5.2    Selection Determination

Much of the ML literature presents raw results and makes a non-scientifically rigorous statement of being "better" by some metric. There are two approaches to developing improved comparisons.

One is to devise better statistical tests to compare two methods when a single test-set is available, first seriously studied by Dietterich (1998) with many follow-up works shortly after (Alpaydin, 1999; Bouckaert, 2003; Bouckaert & Frank, 2004). Different perspectives on this include using one test run to make a conclusion (Dror et al., 2019), or including sources of variation in model performance (e.g., hyper-parameter values) and comparing the distribution of model results (Bouthillier et al., 2019; 2021; Cooper et al., 2021). Others have introduced computational budget for training and parameter tuning as a conditional factor that impacts the conclusion of "best" (Dodge et al., 2019).

The second option is to use multiple datasets to perform a single test of whether one algorithm is better than another (Guerrero Vázquez et al., 2001; Hull, 1994; Pizarro et al., 2002). The use of a non-parametric Wilcoxon test has been found to be efficacious in multiple studies (Demšar, 2006; Benavoli et al., 2016). Dror et al. (2017) extended this to make a conclusion about how many datasets, and which, a method performs better on. Other recent work has proposed the use of meta-analysis methods to make conclusions of a single method tested under multiple conditions (Soboroff, 2018). It is noteworthy that work using multiple datasets to make decisions based on a single evaluation metric implicitly contributes to the Adaptability question, which we explore next.

### 3.6 Adaptability and its Second-Class Status

Adaptability is the study of a different person/team, using the original code, but applying it to their own and different data. Very little work focused on scientific rigor in AI/ML focus on Adaptability. To be clear, many prior works have studied the question of generalization in machine learning – of which there is recent evolution due to the advance of deep learning (Zhang et al., 2017). However generalization assumes some form of intrinsic relationship (usually I.I.D.) between the training and testing distribution. Under Adaptability, there is no direct train/test split to compare. Instead, it is a question of the methodology's effectiveness on an entirely different statistical distribution. Thus our concern is more focused on the practical real-world issues that enable or inhibit a *method* to generalize.

The work we have found can broadly be described as including Adaptability to new datasets, or specialized sub-sets, to better understand the overall behavior and utility of a set of algorithms (Marchesin et al., 2020; Rahmani et al., 2022). The other work that tackles Adaptability is from an HCI perspective in validating a method's utility as population preferences evolve (Roy et al., 2022).

Though it has not been presented as a part of the literature on scientific rigor, considerable effort in the Adaptability question has been advanced by Decision Trees based literature. In particular, the long-standing effectiveness of tree ensembles has led to numerous studies investigating the persistent efficacy of tree ensembles (Grinsztajn et al., 2022; Wainberg et al., 2016). Despite little work on the adaptability question, we make note that many works in Model Selection make use of the adaptability argument as a component of their study or an otherwise latent concern.

### 3.7 Label & Data Quality

Label & Data Quality is focused with the reliability of data and label acquisition, error rates, and working to understand how they occur, detect them, or work around them. Many works today are identifying these issues long after dataset construction, in part due to the high accuracies now being achieved making the errors more pronounced. For example, the process for deriving labels of ImageNet had rules incongruent with the data's nature (e.g., assuming only one class is present) and error-prone steps in the labeling pipeline (Beyer et al., 2020). The issues of label quality also include train/test set leakage (Barz & Denzler, 2020).

Some of the most interesting and insightful research results have come from replicating dataset construction and labeling processes for prior datasets, and then characterizing and discovering why differences in results occur. This includes detection cases where a recreation is implicitly made more challenging than the original dataset (Engstrom et al., 2020).

While there is a long history of research in inferring a single correct label from multiple labelers (Whitehill et al., 2009; Lin et al., 2014; Ratner et al., 2016; Yoshimura et al., 2017; Ratner et al., 2020), this literature is not generally framed as a scientific rigor issue. While these methods have been leveraged in work coming

from a rigor perspective (Beyer et al., 2020), we are not aware of work that bridges a longitudinal study of the replicability of these various label inference procedures.

### 3.8   Meta and Incentives

Very few papers have studied the incentives for scientific rigor. The study of the scientific process itself is often termed meta-science, and when applied to AI/ML research would fall into this category. Such research could include basic studies of incentives, drivers of scientific rigor, and surveys across various AI/ML research domains. Sample studies focused on such drivers as the rate of data and code sharing over Computational Linguistics (Wieling et al., 2018) and the use of statistical testing (Dror et al., 2018). Related work has found that sharing code and producing Replicable research correlate with higher citations (Raff, 2022). Another study focuses on how evaluation and comparison practices evolve throughout the Machine Translation community (Marie et al., 2021). The last work we are aware of challenged the treatment of replicability as a binary "yes/no" question and instead suggests a survival model, where replicability is a function of time/effort (Raff, 2021).

### 3.9   Maintainability

Maintainability is similar to Repeatability, in that we are considered with producing the same results with the original authors (though new users could also occur) using the original code and data. The key difference that distinguishes Maintainability is that time is a factor, as the ability to repeat results degrades over time as nuances of code or software versions change[1]. Maintainability can also deal with the code itself changing over time. The focus on the aspect of maintainability within AI/ML was started by the seminal work of (Sculley et al., 2015). One key area of maintainability deals with adapting known "code-smells" while considering ML-specific concerns and factors that practitioner surveys deem most important (Gesi et al., 2022). Another key area of maintainability is the quality of the results as code itself changes. It has well known that scientific algorithms may produce different results by different, seemingly equivalent, implementations (Hatton, 1993). Multiple studies have found AI/ML is no exception to this history, with large and statistically significant changes in accuracy when using allegedly equivalent algorithms and changing just the implementation, or runtime platform (e.g., GPU hardware) (Coakley et al., 2022; Gundersen et al., 2022; Pham et al., 2020; Zhuang et al., 2021).

## 4   Connections between Rigor Types

Having defined a set of eight rigor types that are being worked on, we further elaborate on our perception of connections between these rigors. In particular, there are direct and indirect relationships, which are summarized in Figure 1 with solid and dashed lines respectively.

### 4.1   Direct Relationships

The most obvious, and intuitive, connections are from Repeatability to Reproducibility to Replicability, as each requires a progressive step of difficulty from the prior. If a single person/team cannot repeat their own experiments, there is no reason to believe a different person with the same code would be able to reproduce those results. Extended further, if they cannot reproduce the results with the original code, there is no special reason to believe that by writing their own code or using different data they would be able to replicate the results.

Less obvious are the interactions between maintainability and repeatability and replicability. First is the two-way relationship between repeatability and maintainability. If an AI/ML system is not repeatable, it cannot be maintainable, as repeatability is the property that we want to maintain. Similarly, if it cannot be maintained it may not be repeatable *over time*. A simple case is the use of Docker to gain repeatability, which is predicated on the repeatability of Docker containers. This assumption is true on short time horizons, but changes in software dependencies, hardware requirements, and eventually deprecation of tools like Docker

---

[1]In software development this notion is often termed "bit rot".

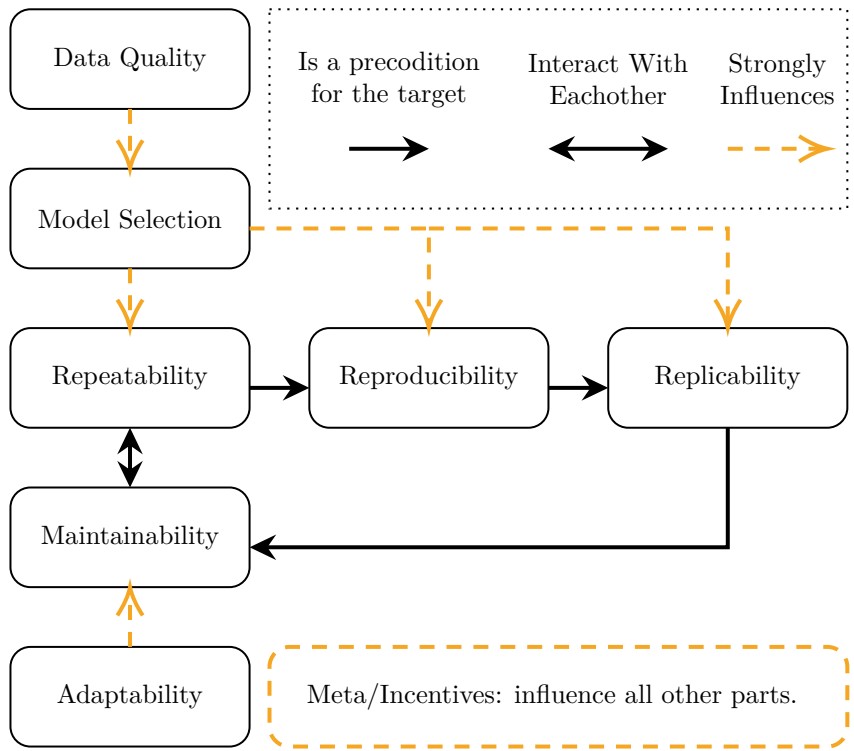

Figure 1: Connections on how rigor types influence each other. Solid lines indicate hard dependencies, while dashed lines show influencing effects.

itself do not make it true in perpetuity. The time-based evolution that maintainability requires then directly implies the replicability of a method. If a system is replicable, meaning the code or data can change as well as the people, it is satisfying the requirement of maintainability over a single point in time. Thus, maintainability involves iterated replicability over time, and instantaneous repeatability at any point in time.

## 4.2 Indirect Relationships

Beyond the general influence of meta and incentives-based rigor having a relationship to all parts of scientific rigor, we can further draw other connections that are of particular note. The most straightforward of these is that of model selection on repeatability, reproducibility, and replicability, each of which will often incorporate the model selection task as part of the motivation for why the proposed work should be used (i.e., it was demonstrated to "be better" than something prior). Thus, by its nature, different approaches to model selection will influence each. For example, the use of Random Search as a hyper-parameter tuning method (Bergstra & Bengio, 2012) is potentially a hindrance to replicability due to higher variance, even if it is easily repeatable and reproducible given the original code with initial seed values for the pseudo-random number generator.

Upstream from this concern is then label and data quality, which will influence what features are selected. This is particularly notable as many datasets reach high accuracies where "errors" in the model's predictions are discovered to be either 1) correct, and that the test data was mislabeled or 2) that the test instance was inherently ambiguous (Barz & Denzler, 2020). This creates a new kind of noise in the selection process and can thus alter conclusions on the merits of what is considered. This is particularly true for the eventual downstream model selection under replicability, where the data in use may be different.

Finally, we note that a method that is adaptable is more likely to be maintainable. The nature of one method being efficacious on many others is the observation that many small details on the implementation

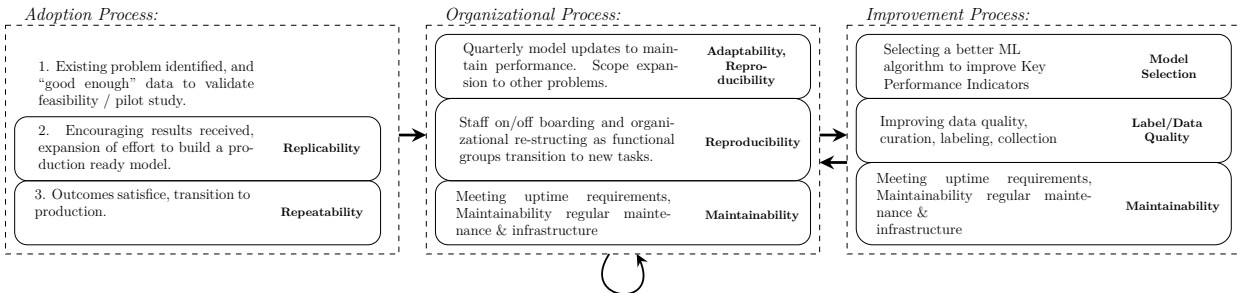

Figure 2: Stylized sketch of how AI/ML may be adopted by an enterprise and involves three overarching processes: adoption, Organizational, and Improvement. Steps in each process relate to different factors of scientific rigor. Businesses begin with an adoption process that determines a satisfactory solution, from which it transitions to being a normal part of the organization's total process. Organization processes may be improved, but are always continuous until deprecated. Maintainability concerns occur in both later stages, Numbered items occur in order, and others are any order.

can vary, while still producing quantitatively similar results, an often observed phenomenon in decision tree literature (Quinlan, 1993; Breiman et al., 1984; Quinlan, 2006; Raff, 2017). This provides some inherent "robustness" to issues that often cause maintainability problems such as changes in low-level libraries like BLAS/LAPACK or new hardware.

## 5    Mismatch Between Academia and Industry

It is important to note the importance (and differences) of scientific rigor for researchers in the scientific community and practitioners in businesses. Within the scientific community, the key goals of reproducing models and experiments are to help (1) build upon existing AI/ML models, (2) compare the proposed model against well-established baselines, and (3) advance knowledge in a particular knowledge base. In contrast, businesses are often concerned with replicating existing (often manual) processes to help establish consistency and reduce costs or gain capability. Thus, for a company to adopt an ML solution to their own problems and data, the approach must be **Replicable** because, fundamentally, they are taking existing research and applying it to a new or existing business problem. *If an ML method is not Replicable, it thus will not be adopted, and the other issues become unimportant.* For this reason we consider Replicability an under-studied issue in the literature given its practical importance.

To highlight this, we sketch a common business lifecycle for the use of machine learning in Figure 2. This can be broken up into three processes:

1. The Adoption process that validates and determines an AI/ML method is valuable and worth using.

2. The Organization processes that constitute the regular and ongoing operations of an enterprise

3. The Improvement process by which a competitive advantage is built. strengthened.

For applications of ML where none existed before (i.e., replacing/augmenting a manual process), issues of reproducibility occur sequentially through these three stages with cycles through the Organizational process. Each scientific rigor question corresponds directly to a real-world need in its own right but may be predicated on other stages having succeeded first. We expand upon these markings below.

The way academia and industry consider **Model Selection** is also important. Current academic research is focused on what is *best*. But in many cases, a company does not care about what is best, but what *satisfices*. Testing an algorithm on many datasets is often used to pick "best", but from a business perspective, they are indirectly demonstrating **Adaptability**. The distinction is often minor academically, but important practically. Any ML method that satisfices by "solving" the business problem is acceptable to the business

and can be adopted. As touched upon in section 3.4, this gives special value to determining *best on a single specific dataset*. This is where a company would potentially derive a competitive advantage, but there are few techniques for robustly selecting a single model with respect to a single dataset beyond the conventional train/test split.

Once a model has been selected, then **Repeatability** and **Reproducibility** come into play. The business unit needs to be able to run the model again, make small tweaks, or transition the solution from IR&D to production and/or service teams. Over time, this becomes more complicated and enters **Maintainability**. The business must plan for multiple years, which also means planning for key personnel turnover. When key people who developed the model in the context of the larger business leave, the ability of new hires to maintain the system in the larger environment may be significantly hampered.

We note as well that in the broader picture of an organization (distinct from the more micro-intrinsic level of the previous section), Adaptability, Maintainability, and Model Selection are all intertwined. Selecting the best model from an accuracy perspective for each problem may lead to some form of competitive advantage, but also produces a maintainability burden: requiring increased documentation of how and why each model was selected, potentially additional and varied hardware and software for inferences, and thus internal overhead. A satisficing model that works for many problems, in turn, reduces this maintenance cost, but at a potential cost to accuracy. Thus, model selection has additional micro and macro-level concerns, of which we are not aware of any formal study. We do note that the XGBoost algorithm is potentially a strong candidate for the study of the macro-level concern due to its support for a wide range of problem types and features (e.g., regression, survival analysis, interpretability, training and inference speed, and broadly its demonstrated adaptability) (Chen & Guestrin, 2016).

In the context of improving a business process, the **Label/Data Quality** factor is likely to be the most important for successful organizations on a regular basis. The value of acquiring more data has long been recognized to produce greater gains than developing new algorithms (Halevy et al., 2009; Banko & Brill, 2001; Domingos, 2012).

## 6 Conclusions

We have synthesized eight current directions in the literature of scientific rigor for machine learning, disentangling them from the commonly repeated moniker of "reproducibility" and thus quantified the proportion of each type as studied today. These rigor types have been further characterized by their interactions/dependencies with each other, and how each is individually important to some part of a business lifecycle in the deployment of AI/ML in the real world.

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

# A  Papers Used

| Citation | Category |
| --- | --- |
| Geiger et al. (2021) | Repeatability |
| Musgrave et al. (2020) | Reproducibility |
| Tatman et al. (2018) | Reproducibility |
| Shepherd et al. (2017) | Reproducibility |
| Raff (2019) | Replicability |
| Zaharia et al. (2018) | Repeatability |
| Bouthillier et al. (2021) | Model Comparison |
| Beyer et al. (2020) | Label Quality |
| Sun et al. (2020) | Model Comparison |
| Dacrema et al. (2019) | Model Comparison |
| Gesi et al. (2022) | Maintainability |
| Paganini & Forde (2020) | Repeatability |
| Dror et al. (2019) | Model Comparison |
| Gundersen et al. (2022) | Maintainability |
| Barz & Denzler (2020) | Label Quality |
| Raff (2022) | Incentives |
| Gardner et al. (2018) | Repeatability |
| Coakley et al. (2022) | Maintainability |
| Marchesin et al. (2020) | Adaptability |
| Eggensperger et al. (2021) | Model Comparison |
| Cooper et al. (2021) | Model Comparison |
| Engstrom et al. (2020) | Label Quality |
| Publio et al. (2018) | Repeatability |
| Matsui & Goya (2022) | Repeatability |
| Gundersen et al. (2018) | Reproducibility |
| Liu et al. (2020) | Reproducibility |
| Zhuang et al. (2021) | Maintainability |
| Dror et al. (2017) | Model Comparison |
| Ahn et al. (2022) | Replicability |
| Forde et al. (2018a) | Reproducibility |
| Drummond (2018) | Repeatability |
| Forde et al. (2018b) | Reproducibility |
| Raff (2021) | Meta |
| Marie et al. (2021) | Meta |
| Dodge et al. (2019) | Model Comparison |
| Wieling et al. (2018) | Meta |
| Gundersen & Kjensmo (2018) | Reproducibility |
| Dror et al. (2018) | Meta |
| Kim et al. (2022) | Model Comparison |
| Bouthillier et al. (2019) | Model Comparison |
| Blalock et al. (2020) | Model Comparison |
| Lu et al. (2023) | Model Comparison |
| Chen et al. (2022) | Repeatability |
| Sun et al. (2022) | Model Comparison |
| Yildiz et al. (2021) | Replicability |
| Lucic et al. (2022) | Repeatability |
| Belz et al. (2020) | Label Quality |
| Lopresti & Nagy (2021) | Meta |
| Moreau et al. (2022) | Model Comparison |
| Arpteg et al. (2018) | Maintainability |

| | |
|---|---|
| Tang et al. (2021) | Maintainability |
| Bogner et al. (2021) | Maintainability |
| Sculley et al. (2015) | Maintainability |
| Kery & Myers (2018) | Repeatability |
| Head et al. (2019) | Repeatability |
| Yang et al. (2021) | Maintainability |
| Rao et al. (2022) | Reproducibility |
| Huang et al. (2022) | Reproducibility |
| Rahmani et al. (2022) | Adaptability |
| Roy et al. (2022) | Adaptability |
| Wang et al. (2022) | Replicability |
| Bates et al. (2021) | Model Comparison |
| Bergmeir et al. (2018) | Model Comparison |
| Varoquaux (2018) | Model Comparison |
| Soboroff (2018) | Model Comparison |

