# OpenReview forum: "Survey of Scientific Rigor Studied in Machine Learning"
_TMLR — Rejected by TMLR_

### Review · Reviewer_iNTD · 2023-07-12

**Summary Of Contributions:**

This paper performs a review of literature studying scientific rigor in machine learning, and puts forth a taxonomy of different aspects of scientific rigor: repeatability, reproducibility, replicability, adaptability, model selection, label/data quality, meta & incentive, and maintainability. The paper discusses which papers have focused on these different types of scientific rigor, as well as conceptual connections between different rigor types. It concludes by explaining that each type of scientific rigor as identified in the paper is important for industry adoption of ML methods.

**Audience:**

No

**Broader Impact Concerns:**

No concerns.

**Claims And Evidence:**

No

**Requested Changes:**

I would like to see this paper make clearer what its contribution and audience are. As it stands, I didn't feel like I learned all that much. The taxonomy is nice, but as I wrote above, it's not clear who should be using it and how. If it could tell me more explicitly what I should take away from it, or what I have learned by reading it, then I believe the quality of the paper would be greatly increased.

**Strengths And Weaknesses:**

Pros:
- Extensive literature review
- ML-specific taxonomy of scientific rigor types
Cons:
- Unclear who the audience is meant to be
- Unclear what conclusions to draw from the paper

It is useful to organize and synthesize an area of research, and I appreciate this paper's attempt to do so for the area of scientific rigor in ML. However, I unfortunately feel that the paper falls short in this attempt. While I agree the types of rigor identified are important, it was not clear to me in reading the paper what to take away beyond that. Who is the audience is meant to be and what is the paper trying to tell that audience?

At times, it feels as though the paper is written for researchers studying scientific rigor in ML: for example, there are many statements in Section 3 along the lines of "this type of rigor is understudied" (the implication being, I suppose, that more people in the area of scientific rigor should study it). At other times, it feels as if the paper is written for a general ML audience: for example, the discussion in Section 5 seems more like a motivation for regular ML researchers to do work that is more rigorous. At the same time, Section 5 also seems like it's aimed at industry practitioners in its attempt to clarify what areas of rigor such practitioners should pay attention to. Because the intended audience is muddied, it's also unclear what the takeaway from the paper is. Is it that more research into scientific rigor is needed? Is it that ML research in general should be more rigorous? Is the taxonomy a guidebook for industry professionals?

It is, of course, ok for a paper to be written for multiple audiences with multiple takeaways, but I find that this paper does not satisfactorily address any of these audiences. If it is written for scientific rigor researchers and the message is that more research is needed, then the paper needs to spend more time discussing what the gaps in the literature are and what the impact would be if they were filled. If it is written for ML researchers at large, then the paper needs to spend more time identifying and discussing approaches that researchers can use to improve the scientific rigor of their papers. And if it is written for industry professionals, then it should provide more of a recipe for how someone in industry can use the taxonomy to decide whether to adopt a method or not.

---

### Review · Reviewer_jJpo · 2023-07-18

**Summary Of Contributions:**

This review paper tackles the "reproducibility crisis" that has cropped up in AI research, with a particular focus on machine learning. The paper follows an ad hoc procedure for identifying relevant papers and analyzes those to identify a set of eight "rigor types" that are often conflated in prior literature. This paper also addresses incentives and the differences between academia vs. industry.

**Audience:**

Yes

**Broader Impact Concerns:**

This paper has the potential to positively impact the community and bring higher standards of scientific rigor. I do not believe this paper requires adding a Broader Impact statement.

**Claims And Evidence:**

Yes

**Requested Changes:**

Please address the weaknesses above. In particular, this paper needs a more rigorous approach to its meta-analysis, improvement in writing quality, and an investigation of RL.

**Strengths And Weaknesses:**

Strengths:
+ This paper addresses an important crisis in the ML community.
+ A discussion on academia vs. industry is included, which is helpful and often missing from many papers.
+ Figure 1 is a nice addition to show the connectivity of the aspects discussed in this paper.
+ The paper defines a helpful taxonomy and breakdown of types of "reproducibility"


Weaknesses:
-It is too bold of a claim to say that one can analyze just 66 papers of the tens of thousands of papers published sense 2017 and say that this review of 66 papers reflects "the focus of the community at large."
-This paper's review is quite biased, relying on "seminal papers known to us." I have not encountered a review paper before that relies on the authors' memory rather than a systematic, *reproducible* algorithm for identifying papers. This is ironic given the authors focus on reproducibility in ML, having seeming ignored reproducibility practices for meta-analyses (e.g., https://tropmedhealth.biomedcentral.com/articles/10.1186/s41182-019-0165-6). Similarly, "subjective exhaustive review" is not appropriate for a review paper. One must list explicit criteria. The criteria listed at the end of Section 2 are one the right track, but the approach as a whole is inadequate and biased.
-This paper is quite short, especially for a review. Perhaps this could be due to the limited review of papers (only 66 since 2017), which could be dealt with using an unbiased or longer-horizon meta-analysis. I was expecting there to be more of a set of positive examples or design guidelines included.
- I recommend that each section have clear, explicit definition, a review of relevant papers on the topic (both positive and negative points), and then a recommendation.
- The paper essentially ignore reinforcement learning (only one citation has this in the title), which is perhaps the most egregious violator of reproducibility, though the field of RL is improving with stable baselines, the Open AI Gym/Mujoco, etc. A further discussion of RL is warranted.

In addition to responding to the weaknesses mentioned above the writing needs to be much improved. For example, just from the first page:
-Stay consistent with numbering. Instead if using '8' or 'eight' interchangeably, pick one.
-Our proposal is that these eight rigor types are...present a model" is not grammatically correct.
-"To encourage more to study" -- more what? more people? More effort?
-AI and ML seems to be represented as distinct communities; however, that is not quite accurate. ML is a subset of AI.
-"is justified, and increasingly" should not have a comma
-"and Replicability which we find useful" should have a comma before "which" as the following is a nonrestrictive clause.
-"We will expand...Our contribution categorized" -- switching between verb tenses
-"section [i]" should be "Section [i]"

---

### Review · Reviewer_Lh6C · 2023-08-15

**Summary Of Contributions:**

The paper is a survey of 66 papers on the topic of scientific rigor in AI and Machine Learning.  The authors argue that these papers can be divided into eight categories: reproducibility, repeatability, replicability, adaptability, model selection, label/data quality, meta & incentive, and maintainability.  They then propose a means of integrating these topics into a cohesive whole so that the reader may understand how these topics relate.  They close by noting the differences in viewpoints between academia and industry on the topic of scientific rigor.

**Audience:**

Yes

**Broader Impact Concerns:**

There are no boarder impacts concerns.

**Claims And Evidence:**

Yes

**Requested Changes:**

Per Weakness 1:

Section 2 has notable limitations, in that the authors assume that this list of papers defines seminal works on scientific rigor in ML.  As noted, they leave out additional papers that are not specifically focused on reproducibility, but a broader set of topics with regards to how ML researchers conduct their work.  Scoping the paper this way by focusing primarily on reproducibility to pose the initial question of the state of scientific rigor in ML, has notable limitations.  I would be less concerned if the researchers did not claim that one can understand the state of scientific rigor research in ML by focusing on reproducibility as a start.

There are two possible ways to remedy this:

- the authors could soften the language around scientific rigor, instead reorienting the titling and replacing wording of "scientific rigor" with "reproducibility" or a related term that indicates that work such as Lipton and Steinhardt ( https://arxiv.org/abs/1807.03341 ) and Sculley et al ( https://openreview.net/forum?id=rJWF0Fywf  ) are out of scope

- the authors could include Lipton and Steinhardt ( https://arxiv.org/abs/1807.03341 ) and Sculley et al ( https://openreview.net/forum?id=rJWF0Fywf  )  in their initial papers and update the paper accordingly

The first option seems simpler to me, and would be sufficient.

Per Weakness 2:

If the authors could provide additional discussion not just synthesizing the calls to action of the papers e.g. adaptability, replicability etc. but also providing a overview or more detailed discussions of the the motivating problems across papers, this would be helpful.

Per weaknesses 3, 4, and 5:

These weaknesses are a limitation of the scope of the methodology defined by the authors.  A discussion of this limitation could suffice, but I think that given the increasing current discussion of these topics, not mentioning them is a notable limitation.  Again, making more narrow claims that this paper is with respect to "reproducibility research" and not "scientific rigor" more broadly could help, but I think that this limitation does suggest that the method of selecting papers does not provide a fully timely and topical set of papers for the reader.  To remedy this, the authors could scrap this method of selecting papers to review entirely, or could provide a limitations discussion of important topics that are out of scope.  This limitation section of important topics out of scope would also alleviate concerns with weaknesses 1 and 2 if these weaknesses are addressed here as well.

**Strengths And Weaknesses:**

Strengths:

The paper provides a broad overview on papers in that highlight the need for increased scientific rigor in the field.

The paper notes differences in topic of emphasis between papers.

The paper provides a means of understanding relationships between these papers.

The paper further emphasizes the need for increased understanding of how research can be improved.

Weaknesses:

1 - Overall scoping - reliance on reproducibility

Titling the paper "Survey of Scientific Rigor Studied in Machine Learning" is rather broad, and excludes important research related to this topic that are not directly related to reproducibility.

The authors use calls for reproducibility as the motivation and the scope of the paper and initially equate scientific rigor with reproducible science.  As noted in the topics of scientific rigor introduced by the authors, reproducibility is not the only concern with regards to scientific rigor in ML.

If we assume that reproducibility is not the only concern, it may be useful to include two additional works. These early works calling for increased scientific rigor in the field both highlight literature that notes problems caused by a lack of rigor and proposes solutions through better methodological practices: Lipton and Steinhardt ( https://arxiv.org/abs/1807.03341 ) and Sculley et al ( https://openreview.net/forum?id=rJWF0Fywf  ).  Given their number of citations, I would consider adding these to the scope of six starting papers.

Moreover, by claiming the title of scientific rigor, the authors do not include in scope papers whose goal is to better understand machine learning findings through rigorous research e.g. Zhang et al  ( https://arxiv.org/abs/1611.03530 ).  As a result, the definition of scientific rigor is a bit misleading, given that one could assume scientific rigor refers to a broad set of practices, not just reproducible science, or the use of scientific rigor to understand ML methods.

I'm uncertain about the underlying premise of the authors that we can use the perception of a reproducibility crisis in ML as a motivator to do a survey of scientific rigor in ML.  This methods leave out a number of important topics in the ML and rigor literature and have some limitations, which are mentioned below.

1a - How the six papers were decided could be further motivated.  The exclusion of papers such as Henderson et al ( https://arxiv.org/abs/1709.06560 ) or Lipton and Steinhardt ( https://arxiv.org/abs/1807.03341 ) and Sculley et al ( https://openreview.net/forum?id=rJWF0Fywf  ) is not clear.

2 - Lack of discussion of technical problems caused by lack of rigor

The introduction notes this problem and then quickly presumes that the problem is considered justified.  Moreover 3.2.2 only discusses a single example paper. To me, this assumption is a bit strong and could be further strengthened with an explanation of the particularities of the need for rigor in the field and the challenges caused by a supposed lack of rigor. Perhaps the authors consider this less important, however I am not certain that the eight topics can be contextualized without an understanding of the specific technical problems noted by these papers:

For example, Moreau et al, Cooper et al, are both concerned with the question of benchmarking optimizers.  This is an important question for both industry and research.  These papers cite Wilson et al (https://arxiv.org/abs/1705.08292 ), who calls into question the usefulness of adaptive gradient optimizers such as Adam when compared to simpler methods such as SGD.

A more notable example is Henderson et al ( https://arxiv.org/abs/1709.06560 ) which is often cited as an example of the reproducibility crisis in ML.  Given the number of citations of the paper, it is a surprise that this is not included.

Many of these papers emphasize why a particular aspect of scientific rigor is important, and the consequences of this lack of rigor. Without discussion of the particular technical challenges posed by a lack of scientific rigor in the field, and a discussion of papers such as Wilson et al that call in to question specific instances of that possible lack of rigor, I am concerned that the paper does not sufficiently distinguish the particularities of AI research as opposed to any other technical field that requires rigorous methodology.


3 - Lack of discussion of the "foundation model" setting

While the selection of papers is comprehensive, the role of large scale settings, in which a researcher or engineer may not even have access to the data or model but only an API in not actively discussed.  Given the lack of transparency of these models, and their increasing importance to research and industry, it seems odd to not mention this topic.  Discussions such as Rogers ( https://hackingsemantics.xyz/2023/closed-baselines/ ) seem relevant and related, but are not discussed.  Perhaps this is due to the scoping of the literature review, but this seems to be a notable blind spot in the discussion.

4 - Discussion of industry without discussion of the auditing literature

The discussion of industrial uses of scientific rigor appear to presume the process for rigor happens internally within an organization.  However, within both the AI safety and fairness community exists literature regarding external evaluation of models (eg https://arxiv.org/abs/2305.15324 and  https://doi.org/10.1145/3306618.3314244 ) .  Again, the connection between scientific rigor and the consequences of a lack of rigor could be further discussed to explain how and if these topics relate.

5 - Discussion of the documentation literature

Limiting the scope to a literature review of these 6 papers misses works on documentation and ML systems.  Examples include https://arxiv.org/abs/1810.03993 , https://arxiv.org/abs/1803.09010 , and https://arxiv.org/abs/2204.10817 These papers seem relevant to the topics identified by the authors, yet are not discussed.

---

### Decision · Action_Editor_3Sy6 · 2023-12-08

**Recommendation:** Reject

**Comment:**

The paper presents a survey on scientific rigor studies, which is an important topic for TMLR. However, we decided to reject the papers based on the following reasons:
- All reviewers think the papers selected in this review may not be conducted in a systematic way. As a result, this paper lacks several important directions in scientific rigors. See concrete suggestions by Reviewer Lh6C (weakness 1) and Reviewer jJpo.
- As a review paper, this paper can benefit from better organization and a more systematic review. We suggest the authors state clear the definitions and scope for each section and organize the discussions in a better way.
- The paper lacks discussion about some important areas, such as reinforcement learning and foundation models. Although it is not necessary to review all the areas in depth, it is worthwhile mentioning those areas briefly.
- As a review paper, we believe the paper can benefit from discussing the potential problems of the studies and future directions, as well as give clear take-away messages.

**Audience:**

Although some individuals in TMLR's audience may be interested in this paper, the survey conducted in this paper is relatively narrow. For example, there's a lack of discussion on foundation models and potential problems caused by rigor.

**Claims And Evidence:**

This paper aims to review the scientific rigor studied in machine learning. However, as most of the reviewers pointed out, the selected papers and topics reviewed in this paper are biased. Therefore, we suggest the authors conduct a more systematic review of the literature to make it more convincing and with clear evidence.

**Resubmission Of Major Revision:**

The authors may consider submitting a major revision at a later time.